# Effect of Mifepristone on Migration and Proliferation of Oral Cancer Cells

**DOI:** 10.3390/ijms25168777

**Published:** 2024-08-12

**Authors:** Anem Iftikhar, Simon Shepherd, Sarah Jones, Ian Ellis

**Affiliations:** School of Dentistry, University of Dundee, Dundee DD1 4HR, UK; a.iftikhar@dundee.ac.uk (A.I.); s.d.shepherd@dundee.ac.uk (S.S.); s.j.jones@dundee.ac.uk (S.J.)

**Keywords:** glucocorticoid receptor, head and neck cancer, mifepristone, oral cancer, PI3K/Akt signalling pathway, MAPK signalling pathway, cell migration

## Abstract

Glucocorticoid receptor (GR) overexpression has been linked to increased tumour aggressiveness and treatment resistance. GR antagonists have been shown to enhance treatment effectiveness. Emerging research has investigated mifepristone, a GR antagonist, as an anticancer agent with limited research in the context of oral cancer. This study investigated the effect of mifepristone at micromolar (µM) concentrations of 1, 5, 10 and 20 on the proliferation and migration of oral cancer cells, at 24 and 48 h. Scratch and scatter assays were utilised to assess cell migration, MTT assays were used to measure cell proliferation, Western blotting was used to investigate the expression of GR and the activation of underlying Phosphoinositide 3-kinase (PI3K)/protein kinase B (Akt) and mitogen-activated protein kinase (MAPK) signalling pathways, and immunofluorescence (IF) was used to determine the localisation of proteins in HaCaT (immortalised human skin keratinocytes), TYS (oral adeno squamous cell carcinoma), and SAS-H1 cells (squamous cell carcinoma of human tongue). Mifepristone resulted in a dose-dependent reduction in the proliferation of HaCaT, TYS, and SAS-H1 cells. Mifepristone at a concentration of 20 µM effectively reduced collective migration and scattering of oral cancer cells, consistent with the suppression of the PI3K-Akt and MAPK signalling pathways, and reduced expression of N-Cadherin. An elongated cell morphology was, however, observed, which may be linked to the localisation pattern of E-Cadherin in response to mifepristone. Overall, this study found that a high concentration of mifepristone was effective in the suppression of migration and proliferation of oral cancer cells via the inhibition of PI3K-Akt and MAPK signalling pathways. Further investigation is needed to define its impact on epithelial–mesenchymal transition (EMT) markers.

## 1. Introduction

Mifepristone was synthesised in 1982, originally as a GR antagonist. Later, it was identified as a progesterone receptor (PR) blocker at low doses. The binding affinities of mifepristone to PR and GR are higher than that of their ligands, progesterone and cortisol, respectively. Due to its anti-progesterone activity, it was primarily recognised as a non-invasive abortifacient [1] and was added to the list of essential drugs in reproductive health in 2005 by the WHO [2]. 

Broadened beyond an abortifacient, mifepristone has been investigated as a pharmacological agent in managing alcohol-use disorders, and stress-related comorbidities that involve elevated cortisol levels [3]. The application of mifepristone in oncology has gained significant attention in the past few decades, due to its anti-proliferative effects on cancer cells [4]. The literature has reviewed the transdisciplinary similarities between embryonic implantation into the endometrium and vascular invasion of tumour cells resulting in research into mifepristone as an effective drug for metastasis prevention [1,5]. 

The anticancer role of mifepristone has been investigated in cell lines [6,7,8], animal models, and clinical trials [9,10,11,12]. The initial studies examining mifepristone as an anticancer agent were focused on reproductive tumours [13] or hormone-sensitive tumours because of its then widely known and applied anti-progestin activity [14] in reproductive health. Therefore, studies focused on breast, ovarian and prostate cancers. Some clinical trials investigating the impact of mifepristone have been unsuccessful [15]. This has been attributed to the complex crosstalk and interaction of steroid receptors in these cancers, and, hence, the results of blocking one receptor depend upon the presence and absence of other steroid receptors in that cancer type. Preclinical studies investigated the anticancer properties of mifepristone in bone, nervous tissue, osteosarcoma, and non-small cell lung carcinoma (NSCLC), as its anti-tumorigenic effects were understood to occur irrespective of PR expression [16].

Oral cancer is categorised within the classification of head and neck cancer (HNC), the sixth most common cancer worldwide, with 870,000 cases and 440,000 deaths in 2020 [17]. The incidence of HNC continues to increase, and it is predicted to reach 1.08 million new cases per year by 2030 [18]. Patients are often diagnosed at a late stage due to a lack of both early symptoms and efficient screening strategies [19]. Despite progress in therapies, including targeted therapy and checkpoint inhibitors [20], long-term survival in HNC remains poor, with approximately fifty percent of cases ending in tumour recurrence, metastasis, or death within five years [21]. 

Amongst the multiple risk factors for HNC are age, ethnicity, socioeconomic status, infections, genetic alterations, and environmental factors. However, lifestyle risk factors, including alcohol and smoking, are undeniably major contributors in the development and progression of the disease [22]. An increased prevalence of stress has been reported in patients with HNC [23]. Alcohol-use disorders and stress-related comorbidities are characterised by dysregulation of the hypothalamus pituitary adrenal axis (HPA) axis, involving the role of GR [24]. Previously, we reviewed how stress acts as an upstream factor in the aetiology of HNC via the HPA axis, cortisol release, and GR signalling [25]. In addition to being increasingly investigated as an anticancer agent in other cancer types, mifepristone’s potential involvement in stress-related comorbidities and reduction in alcohol craving due to GR antagonism further highlight the significance of investigating its role as an anticancer agent in HNC. 

Cell proliferation and migration are the key hallmarks of cancer and are crucial to metastasis [26]. Cells can adopt distinct modes of migration, including single cell and collective cell migration [27,28]. Single cells move as individual entities that are not connected to neighbouring cells. In collective migration, cells retain cell–cell junctions and move in clusters or in strands consisting of the leader cell at the tip of the group and follower cells [29]. Cell migration is in turn, influenced by a transition in cellular morphology from epithelial to mesenchymal, a process known as EMT [30]. Dysregulation of the PI3K-Akt and MAPK signalling pathways has been linked to increased cell proliferation and cell migration in HNC. While some studies in HNC have investigated the effect of mifepristone on cell proliferation; however, in the context of oral cancer, the role of mifepristone is not widely documented, especially in relation to cell migration. Seeking to address this gap, the objectives of this study were to investigate how a range of concentrations of mifepristone affected the migration and proliferation of oral cancer cells, the underlying PI3K-Akt and MAPK signalling pathways, and EMT markers (E-Cadherin and N-Cadherin) in oral cancer cells.

## 2. Results

### 2.1. TYS and SAS-H1 Cells Showed Increased Expression of GR with a Pronounced Nuclear Presence in Comparison to HaCaT Cells

The expression of GR was investigated prior to treating the cells with mifepristone to determine whether HaCaT, TYS, and SAS-H1 cells expressed GR and also whether cancer cells differed from the normal cells in terms of GR expression and localisation. All three cell lines expressed GR as depicted in the Western blot image in Figure 1A. Full blot images are presented in the Appendix A. Oral cancer cells, namely TYS and SAS-H1, showed an increased expression of GR in comparison to the normal cells, as in Figure 1A. The highest expression of GR was observed in the TYS cells, as in Figure 1A. Localisation of GR was nucleocytoplasmic with a pronounced nuclear presence in TYS and SAS-H1 cells, as in Figure 1B(ix,xiv), in comparison to the HaCaT cells, as in Figure 1B(iv), which showed a cytoplasmic localisation of GR.

### 2.2. Effect of Mifepristone on Expression and Localisation of the GR

To determine the effect of mifepristone on the GR expression and localisation, the HaCaT, TYS, and SAS-H1 cells were treated with increasing concentrations of mifepristone (Table 1). Cells incubated with SFM were used as a control. Treatment of HaCaT and SAS-H1 cells with all concentrations of mifepristone resulted in a decreased expression of GR, when compared to the control at 24 and 48 h Figure 2A, (full blot images are presented in the Appendix A). Treatment of the TYS cells with 20 µM mifepristone resulted in decreased expression of GR in comparison to the control at 24 h Figure 2A. In contrast to weak staining of GR and cytoplasmic localisation in HaCaT cells, a strong nucleocytoplasmic localisation was observed in SAS-H1 and TYS control cultures. Treatment of cells with 20 µM mifepristone resulted in complete nuclear translocation of GR in HaCaT, TYS, and SAS-H1 cells at both time points, as in Figure 2B.

### 2.3. Mifepristone Resulted in Reduced Cell Proliferation in a Dose-Dependent Manner in All Cell Lines

Treatment of HaCaT, TYS, and SAS-H1cells with mifepristone resulted in a dose-dependent decrease in proliferation, in comparison to the control, at both 24 and 48 h, as in Figure 3. A 40–50% decrease in proliferation was observed at 24 h in response to treatment with mifepristone concentrations of 10 µM in the HaCaT cells, as in Figure 3A, and 20 µM in the oral cancer cells, as in Figure 3B,C, in comparison to incubation with SFM.

### 2.4. High Concentrations of Mifepristone Resulted in Decreased Collective Cell Migration and Compact Colonies of Oral Cancer Cells

Following the change in expression and localisation of GR in response to mifepristone, the effect of mifepristone was investigated on collective cell migration and cell scatter. The scratch assay photomicrographs for HaCaT, TYS, and SAS-H1 cells (Figure 4A) were captured at time point 0 (T = 0) and after 24 h (T = 24 h) of treatment with mifepristone (1, 5, 10 and 20 µM), to observe the effect on collective cell migration. HaCaT cells did not migrate in the control cultures and, therefore, no effect of mifepristone was observed. An effective reduction in collective migration of TYS and SAS-H1 cells was only observed in response to treatment with 20 µM mifepristone in comparison to the control, as in Figure 4A(I,II)). The scatter assay photomicrographs of HaCaT, TYS, and SAS-H1 cells in Figure 4B were taken at time point 0 (T = 0) to observe the colonies, followed by 24 h (T = 24 h) and 48 h (T = 48 h) of treatment with experimental conditions, to observe the effect of mifepristone on cell migration. Cell migration was observed in TYS as shown in Figure 4B(ii,vii) and in SAS-H1 as shown in Figure 4B(ii,vii) control cultures (green arrows indicating scattered cells), but not in HaCaT control cultures. As the concentration of mifepristone increased, colonies of TYS and SAS-H1 cells appeared to be more compact in comparison to the control. However, within the compacted colonies, elongated morphology of TYS (Figure 4B(x,xi)) and SAS-H1 (Figure 4 B(vi)) cells was observed, more prominent in response to 20 µM mifepristone as indicated by blue arrows in Figure 4B.

### 2.5. Effect of Mifepristone on the PI3K-Akt Signalling Pathway

Treatment of HaCaT, TYS, and SAS-H1 cells with concentrations of 10 and 20 µM mifepristone resulted in decreased levels of pAkt T308 and pAkt S473 in comparison to the control cultures, as in Figure 5A(i,ii), (full blot images are presented in the Appendix A). Cells in response to treatment with mifepristone concentrations of 1 and 5 µM showed similar or higher levels of phosphorylated Akt T308 or S473, when compared to control cultures at one of the time points, resulting in ineffective inhibition of the Akt signalling pathway in the oral cancer cells. Treatment of oral cancer cells, TYS and SAS-H1, with 20 µM mifepristone resulted in markedly reduced levels of pAkt T308 and pAkt S473, at 24 and 48 h with weak cytoplasmic localisation, as in Figure 5B(ii,iii), in comparison to control cells incubated with SFM that showed a nucleocytoplasmic/membranous localisation. Phosphorylated Akt T308 and pAkt S473 were not detected in HaCaT cells, as in Figure 5B(i).

### 2.6. Effect of Mifepristone on the Phospho p44/42 MAPK Signalling Pathway

Treatment of HaCaT, TYS, and SAS-H1 cells with 20 µM mifepristone in comparison to control cultures incubated with SFM resulted in decreased levels of phospho p44/42 MAPK at 24 and 48 h, as in Figure 6A, (full blot images are presented in the Appendix A). Cells treated with other concentrations of mifepristone showed levels of phospho p44/42 MAPK that were either increased or comparable to the levels found in control cultures at one of the two time points. A strong nucleocytoplasmic staining for phospho p44/42 MAPK was observed in TYS and SAS-H1 control cultures, as in Figure 6B. In contrast, few cells showed staining for phospho p44/42 MAPK when treated with mifepristone, indicating a weak nucleocytoplasmic localisation in TYS cells and cytoplasmic localisation in SAS-H1 cells, as in Figure 6B. Phospho p44/42 MAPK was not detected in HaCaT control cultures or in response to mifepristone in the immunofluorescence assay (Figure 6B).

### 2.7. Effect of Mifepristone on EMT Markers

The findings of compacted colonies along with the elongated cell morphology of oral cancer cells in response to increasing concentrations of mifepristone led to the investigation of EMT markers. In HaCaT cells, the expression of E-Cadherin in response to low concentrations of mifepristone was similar to control cultures incubated with SFM. The expression was observed to decrease in the HaCaT cells treated with 10 and 20 µM mifepristone from 24 to 48 h, Figure 7A(i) (full blot images presented in the Appendix A, in Appendix A). The localisation of E-Cadherin shifted from membranous to cytoplasmic and nuclear between 24 and 48 h in cells treated with 20 µM mifepristone, as in Figure 7B. A simultaneous decrease in N-Cadherin was observed in HaCaT cells treated with 20 µM mifepristone. In response to other concentrations, the expression of N-Cadherin was similar or higher compared to the control, as in Figure 7A(ii). The expression of E-Cadherin in TYS and SAS-H1 cells treated with 20 µM mifepristone was observed to increase from 24 to 48 h or remain similar when compared to control cultures, as in Figure 7A(i). In terms of localisation (Figure 7B), E-Cadherin was downregulated where cells were further apart and membranous where cells were in colonies, in both TYS and SAS-H1 control cultures. Mifepristone-treated cells showed nucleocytoplasmic expression along with membranous presence of E-Cadherin. The expression of N-Cadherin in TYS cells was observed to decrease in response to all concentrations of mifepristone at 24 and 48 h except 10 µM, as in Figure 7A(i). A dose-dependent decrease in N-Cadherin was observed in SAS-H1 cells treated with mifepristone, as in Figure 7A(i). N-Cadherin was not detected in IF, in response to mifepristone-treated TYS and SAS-H1 cells, as in Figure 7B. 

## 3. Discussion

The use of mifepristone as a GR antagonist has been reported in solid tumours as an attractive anticancer agent [4] by inhibiting growth and proliferation [16], suppressing invasive metastatic potential [31], and reversing resistance to chemotherapeutic agents [32]. In contrast, it has also been demonstrated that mifepristone exhibits agonistic properties which promote growth in some tumours. The agonist or antagonist properties of mifepristone involve complex mechanisms influenced by many factors and are still being researched. These factors include the type and expression level of hormone receptors, the function of these receptors in a specific cancer type, their interaction with each other, and the dosage of mifepristone [15]. 

The expression of GR has been reported in head and neck tumour samples and cell lines [33,34,35,36,37]. Recent HNC studies demonstrated the role of glucocorticoids via GR in reducing the efficacy of chemotherapeutic agents [38,39]. The role of mifepristone has been investigated in terms of its anti-proliferative effects and reversing the resistance to chemotherapy. However, the activity of mifepristone has not been investigated in the context of oral cancer cell migration and the related underlying signalling pathways. This study utilised a range of mifepristone concentrations to determine its effect on oral cancer cell migration, proliferation, underlying Akt and MAPK signalling pathways, and EMT markers. Additionally, the significance of investigating the effect of mifepristone in oral cancer also stems from the strong relationship between alcohol consumption [40] and oral cancer, as well as the increased prevalence of stress in patients with oral cancer [41], and how mifepristone as a GR antagonist may potentially contribute to suppression of alcohol craving. 

The basal expression of GR was observed to be higher in the oral cancer cells (TYS and SAS-H1) than in the normal cells (HaCaT). Similar to previous studies [35,36,37], a nucleocytoplasmic localisation was observed in control cultures. The addition of mifepristone to cells resulted in nuclear translocation of GR, consistent with that previously reported in a mouse pituitary gland tumour cell line [42]. In an unbound state, GR resides in the cytoplasm, and upon ligand (cortisol) binding, the receptor undergoes a conformational change resulting in nuclear translocation. In the nucleus, cortisol–GR complexes bind to glucocorticoid receptor elements (GREs) on the genes, influencing gene transcription. Mifepristone, with a higher binding affinity to GR than cortisol, also prompts nuclear translocation. However, inside the nucleus, recruitment of different coactivators and co-suppressors leads to different secondary effects on gene transcription than those in response to the ligand [43]. The findings of elevated GR levels in oral cancer cells led to a subsequent investigation into the effect of mifepristone treatment on cell migration and proliferation. 

The addition of 20 µM mifepristone to oral cancer cells resulted in an effective reduction in collective cell migration when compared to the control. Similar findings were reported by a study on ovarian cancer cell lines, in which mifepristone reduced gap closure [44], and this was linked to an inhibitory effect of mifepristone on the SDF-1/CXCR-4 axis with downstream suppression of the PI3K-Akt and MAPK signalling pathways. A dose-dependent inhibition of collective cell migration using metapristone, a metabolite of mifepristone that was also reported in breast cancer cells [45]. However, the authors reported a 43% inhibition in gap closure at a concentration of 75 µM, in contrast to an effective inhibition observed at 20 µM in this study. This could indicate a cell line-dependent effect of mifepristone and, hence, the difference in the reported effectiveness of varying concentrations. 

Treatment of TYS and SAS-H1 cells with a mifepristone concentration of 20 µM improved colony compactness from 24 to 48 h, and no individual cells were observed, in comparison to cells incubated with SFM. However, a dose-dependent elongated morphology was observed that became more prominent with increasing concentrations of mifepristone. Comparable findings in cell lines from breast, ovary, prostate, and nervous tissue cancers [7] appear to confirm a time-dependent change in the morphology of cells treated with mifepristone, and cells were observed to have an elongated spindle-shaped appearance over 24 to 48 h. Although this morphology is associated with increased migration, in the context of mifepristone it has been linked to dysregulated cytoskeletal structure, including both actin filaments and alpha tubulin. This reduces the adhesive capacity of cells and seeding at the secondary site, which are essential to migration [46]. 

However, contrary to our findings of elongated morphology, a rounded and epithelial phenotype was reported in ovarian [44] and breast [45] cancer cells. These findings were linked to increased E-Cadherin expression, as reported in melanoma cells [47] and endometrial cancer cells [48]. An upregulation of E-Cadherin and a downregulation of vimentin in relation to the epithelial phenotype were also reported in response to mifepristone in breast cancer cells [45]. Our findings demonstrated that the expression of E-Cadherin in oral cancer cells was equivalent to or greater than that of control cultures at 48 h, following treatment with 20 µM mifepristone. However, N-Cadherin was decreased at both time points. The elongated morphology of cells could be attributed to the localisation of E-Cadherin in response to mifepristone, since the functionality of E-Cadherin is dependent not only on its expression, but also its localisation. Elevated E-Cadherin that is delocalised from the membrane is regarded as aberrant and non-functional, and is reported to be associated with heightened invasiveness [49]. In our experiments, mifepristone resulted in membranous and nucleocytoplasmic expression of E-cadherin. This was in contrast to an entirely membranous expression reported in breast cancer cells [45] or the complete downregulation or delocalisation found in response to the SFM (control), which was consistent with scattered cells and collective cell migration in control cultures. Despite cytoplasmic expression, simultaneous retention of E-Cadherin at the membrane and decreased expression of mesenchymal marker, N-Cadherin, may explain the suppression in collective cell migration and compact colonies in response to mifepristone. This observation warrants additional research, as few studies report the localisation of E-Cadherin in response to mifepristone, especially in oral or head and neck cancers. Inhibition of cell migration by mifepristone has also been reported in breast, prostate, ovary and glioma cancer cell lines. Using the scratch and Boyden chamber assays, the authors found a decrease in the migratory capacity of all cell lines [50]. However, Ritch and colleagues used a different approach to cell treatment. In their study, cells were exposed to mifepristone for 72 h prior to creating the gap for the gap closure assay, which may have increased the effectiveness of mifepristone. This study did not involve pre-treatment; however, the results still indicated an effective inhibition of gap closure in the assay.

Most studies on the anticancer effect of mifepristone have focused on its anti-proliferative effect and linked it to an upregulation in cyclin-dependent kinase inhibitors and arresting the transition from the G1 to S phase of the cell cycle [51]. In the cell proliferation assay, mifepristone resulted in decreased cell proliferation in a concentration-dependent manner in comparison to the control. Similar findings have been reported by studies investigating the effect of mifepristone on cell proliferation and growth in uveal melanoma [52], ovarian cancers [53], and neuroblastoma [54]. The inhibitory effect of mifepristone on cell proliferation was observed in all three cell lines, in contrast to a differential effect according to GR expression reported in salivary duct carcinoma cell lines. The authors in this study demonstrated that cell lines with high GR expression showed growth inhibition in response to mifepristone, whereas inhibition was not observed in cell lines that exhibited a low GR expression [55]. 

PI3K-Akt and MAPK signalling pathways play a central role in the tumour biology of HNC. The PI3K/Akt pathway is documented to be the most commonly dysregulated pathway in HNC and is active in 90% of HNSCC cases. Dysregulation of these pathways has been linked to cell migration, proliferation, and survival [19]. Inhibitors targeting these signalling pathways have been developed and investigated in pre-clinical models and clinical trials [56,57]. Pre-clinical studies suggested growth arrest in response to PI3K-Akt and MAPK pathways inhibitors [56,57,58,59]. Clinical trials have not shown encouraging results due to low efficacy, toxicity, and drug resistance [60]. Crosstalk, complex interconnections, and feedback loops between the signalling cascades can lead to hyperactivation of one signalling pathway upon abrogation of another. 

Following the observed effects of mifepristone on cell migration and proliferation, we subsequently examined how mifepristone affected these signalling pathways. Effective blocking of PI3K-Akt and phospho p44/42 MAPK signalling pathways was achieved at 20 µM mifepristone in oral cancer cells, also reported in ovarian cancer, lung cancer [61,62], and endometrial cancer [63]; however, the latter showed the effectiveness of a mifepristone dose over 50 µM, which was too high for our cell lines and resulted in cell death. The findings from the analysis of the signalling pathways aligned with results from the scratch and scatter assays. At lower concentrations of 1 and 5 µM, mifepristone was found to be less effective at blocking either the levels of phosphorylated Akt or phospho p44/42 MAPK in the oral cancer cell lines and accordingly a decrease in cell migration was not observed in scratch or scatter assays. These findings highlight mifepristone’s effectiveness in inhibiting these Akt and MAPK signalling pathways, suggesting that it would be valuable to investigate whether combining mifepristone with inhibitors of these pathways and targeting GR could amplify the effects of these pathway inhibitors.

Glucocorticoid receptors have been linked to aggressive tumours and resistance to treatment [64]. Glucocorticoid receptor antagonists, both selective and non-selective, have been shown to increase the effectiveness of chemotherapy [65]. The success of mifepristone as an anticancer agent is context-dependent, varying across different cancer types. While mifepristone’s anticancer effects are widely reported for other cancer types, there is lack of detailed exploration in head and neck cancer. This study bridges the gap by investigating the effects of mifepristone on cell migration, proliferation, and the underlying signalling pathways in oral cancer cells. However, limitations of this study should be covered by future work that should include the exploration of mifepristone effects in HNC-engineered mice, increasing the number of cell lines studied, and expanding the focus to include additional epithelial markers, such as cytokeratins, Zona Occludens-1 (ZO-1), claudins, occludin, and mesenchymal markers, such as vimentin, and fibronectin. Results from this study demonstrated a dose-dependent reduction in proliferation in all cell lines. Mifepristone resulted in the effective inhibition of collective cell migration and the suppression of scattering of oral cancer cells at high concentrations (20 µM), despite the elongated morphology and cytoplasmic localisation of E-Cadherin. The inhibitory effects of mifepristone were consistent with the suppression of the PI3K-Akt and MAPK signalling pathways and reduced expression of N-Cadherin. However, cytoplasmic localisation of E-Cadherin and its effect in terms of mifepristone usage in oral cancer cell lines should be explored further. Future investigation should focus on additional EMT markers to fully understand the role of mifepristone and also on related signalling pathways to fully elucidate its use as an inhibitor in oral cancer. 

## 4. Materials and Methods

### 4.1. Cell Lines, Culture, and Conditions 

All experiments were performed using a normal cell line (HaCaT) and oral cancer cell lines (TYS, SAS-H1). TYS (oral adeno squamous cell carcinoma) and SAS-H1 (squamous cell carcinoma of the human tongue) were gifted by Dr. Koji Harada, University of Tokushima, Japan. The HaCaT cell line was gifted by Professor SL Schor (late), Dental School, University of Dundee, as in Table 1. Cells were maintained in minimum essential medium (MEM) (SIGMA, St Louis, MO, USA), supplemented with 10% (*v*/*v*) foetal calf serum (FCS) (LIFE TECH/GIBCO) and 200mM glutamine (SIGMA, St Louis, MO, USA), penicillin–streptomycin stabilised (SIGMA, St Louis, MO, USA), and sodium pyruvate (SIGMA, St Louis, MO, USA) at optimal growth conditions of 37 °C and 5% CO_2_. Mifepristone (#M8046, Sigma Aldrich, St Louis, MO, USA) was dissolved in DMSO to make a stock concentration of 20 mM, stored at −20 °C. Mifepristone was prepared in serum-free medium (MEM-SIGMA, St Louis, MO, USA) to achieve working concentrations of 1, 5, 10, and 20 µM for experiments. Serum-free medium (SFM) without mifepristone was used as a control.

### 4.2. Experimental Conditions

The TYS, SAS-H1, and HaCaT cells were treated with the experimental conditions summarised in Table 1.

### 4.3. Scratch Assay

HaCaT, TYS, and SAS-H1 cells were seeded at 5 × 10^5^ cells per 60 mm dish in 10% FCS-MEM and grown in optimal conditions until 90 to 100% confluency was achieved. Following overnight serum starvation, a scratch was made to create a gap in the cell monolayer. The cells were incubated in experimental conditions, summarised in Table 1*,* for 24 h. Images were captured for all experimental conditions at time point 0 (T = 0) and after 24 h (T = 24 h). The change in the gap in response to experimental conditions was observed and compared to the initial gap area. Image J software (Java 1.8.0-345 64-bit) was used to analyse the percentage of gap closure, as described in [66]. The graphical data were presented as a mean of three experiments, with error bars showing the standard deviation. 

### 4.4. Scatter Assay

HaCaT, TYS, and SAS-H1 cells were seeded at 4 × 10^4^ cells per 60 mm and grown in optimal growth conditions. Following the formation of small colonies, cells were serum starved. Experimental conditions were added for 24 and 48 h. Images were taken at the beginning (T = 0) and at the completion of each incubation time point (T = 24 h, T = 48 h). The effect of each concentration of mifepristone was observed on cells in terms of colony compactness, cell-to-cell distance, cell morphology, and the presence of individual cells.

### 4.5. Cell Lysis, SDS-PAGE, and Western Blot

HaCaT, TYS, and SAS-H1 cells were seeded at 1 × 10^5^ cells per 60 mm dish and grown in optimal growth conditions. After serum starvation for 24 h, cells were treated with experimental conditions, including mifepristone at concentrations of 1 µM, 5 µM, 10 µM, and 20 µM for 24 and 48 h. At the completion of a respective incubation, medium was aspirated from the cells, the cells were washed with PBS, and then 0.5 mL/dish RIPA buffer (50 mmol/L Tris HCl, 150 mmol/L NaCl, pH 7.4; 0.1% *w*/*v* SDS, 1% *v*/*v* Triton x-100, 1% *w*/*v* sodium deoxycholate, and 5 mmol/L EDTA) containing protease and phosphatase inhibitors (Roche applied science, Penzberg, Germany) was added. The dishes were then placed on a bed of ice for 10 min. The cells were scraped with pastettes and the lysates were collected in labelled Eppendorf tubes for each experimental condition. The lysates were frozen at −20 °C until they were used for SDS-PAGE and Western blotting, as described previously [67]. Antibody details and concentrations are described in Table 1.

### 4.6. Proliferation Assay (MTT)

HaCaT, TYS, and SAS-H1 cells were seeded in 48-well plates at a density of 2 × 10^4^ cells per well. Following attachment overnight, cells were incubated under experimental conditions (Table 1) for 24 and 48 h. After completion of the incubation times, MTT solution (3(4,5-dimethylthiazolyl-2)2, diphenyltetrazolium bromide) (SIGMA) diluted in serum-free medium (SFM) was added to the wells, and the plates were then incubated for 3 h at 37 °C. The MTT solution was then removed from the wells and DMSO (dimethyl sulfoxide) was added to solubilise the formazan crystals, resulting in a purple-coloured solution. The plate was placed on an orbital shaker for 30 min and read using the OPTIMA plate reader (BMG Lab tech, Offenburg, Germany) at a wavelength of 550 nm and a reference filter of 620 nm. Cell proliferation percentage was calculated using the following equation:(Final OD of sample − OD blank/Final OD serum free − OD blank) × 100.

### 4.7. Immunofluorescence (IF)

Cells were seeded at 4 × 10^4^ cells per 60 mm dish and treated with experimental conditions after a 24 h serum starvation. Following the completion of the respective incubation times, cells were fixed with cold methanol for 20 min followed by 3x PBS washes. Cells were then permeabilised with 0.2% Triton X 100 in PBS for 5 min followed by 3x PBS washes and rings were then drawn around the cells with an Immunopen (Dako, Cambridgeshire, UK), followed by incubation with 5% NGS-PBST (Vector Laboratories, Burlingame, CA, USA) blocking buffer for 1 h. Cells were then washed and incubated with primary antibodies overnight (Table 1). The primary antibody was removed, followed by 3x PBS washes, and cells were then incubated with secondary antibody for one hour at room temperature. This was followed by 3x PBS washes and incubation with 1 µg/mL DAPI (#4083 Cell Signaling Technology, Danvers, MA, USA) for 5 min, to stain the nuclei. DAPI was then removed, cells were rinsed with PBS, and mounting media and cover slips were then placed. Images of cells were then captured using the fluorescence microscope (IX 70) and the digital camera (XM10) at 200× magnification. The DAPI and fluorescent labelled antibody images were merged using Photoshop.

## 5. Conclusions

Mifepristone has attracted significant attention as an anticancer agent due to its anti-proliferative effect and its ability to interfere with signalling pathways essential to cancer cell survival, metastasis, and resistance to chemotherapy. Detailed exploration regarding its impact on oral cancer remains limited. While not directly linked to its anticancer properties, the evidence suggesting that mifepristone could influence behaviour related to alcohol craving further underscores the importance of research in the context of HNC, where alcohol is a known risk factor. 

This study (summarised in Figure 8) highlights the substantial impact of mifepristone in oral cancer cells, demonstrating that high concentrations (20 µM) effectively reduce the collective migration and scattering of oral cancer cells. This effect is in accordance with the suppression of the Akt and MAPK signalling pathways, as well as the decreased expression of N-Cadherin. Despite the observed elongated cell morphology, which may be associated with the localisation pattern of E-Cadherin in response to mifepristone, the findings underscore the efficacy of mifepristone in inhibiting both the migration and proliferation of oral cancer cells (Figure 8). However, further investigations are warranted to explore its impact on EMT markers to fully elucidate its role in HNC.

## Figures and Tables

**Figure 1 ijms-25-08777-f001:**
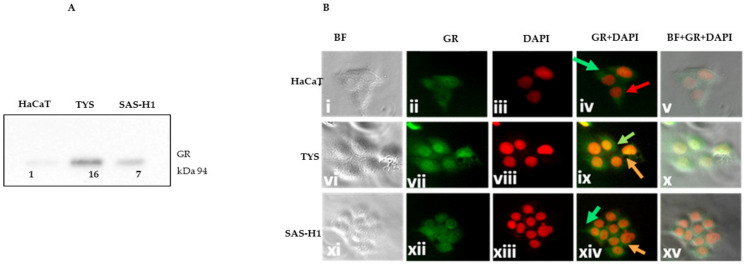
Expression and localisation of the glucocorticoid receptor (GR) in HaCaT, TYS, and SAS-H1 cells. (**A**) HaCaT, TYS, and SAS-H1 cells were seeded at 1 × 10^5^ cells per 60 mm dish and grown in 10% foetal calf serum–minimum essential medium (FCS-MEM) for 24 h. Cells were then lysed, and the lysates were fractionated by SDS-PAGE using 10% acrylamide gels followed by transfer to a PVDF membrane by Western blotting. The membrane was probed using primary antibody for GR (1:1000), followed by incubation with goat anti-rabbit secondary antibody (Table 1). The Western blot was quantified against total protein, and the data are presented as the fold-change in the GR for TYS and SAS-H1 cells compared to HaCaT cells. TYS and SAS-H1 cells showed a higher expression of the GR in comparison to the HaCaT cells. Full blot images are presented in the Appendix A. (**B**) (i–xv) HaCaT, TYS, and SAS-H1 cells were fixed with methanol after incubation with 10% FCS-MEM for 24 h. The fixed cells were analysed for the localisation of the GR by immunofluorescence (IF) with primary and fluorescent labelled secondary antibodies. The HaCaT, TYS, and SAS-H1 cells were observed for localisation of the GR using a fluorescence microscope IX70 and images were captured with a digital camera, XM10. Bright field (BF) images displayed cells, DAPI-stained images indicated the nuclei, and labelled antibody images revealed the GR. BF, GR, and DAPI images were merged using Adobe Photoshop to observe the pattern of localisation of GR in the cytoplasm and/or nucleus of the cells. Red arrow indicates the nucleus. Green arrows show localisation in the cytoplasm, orange arrows show localisation in the nucleus. Experiments were repeated three times. A representative experiment is shown.

**Figure 2 ijms-25-08777-f002:**
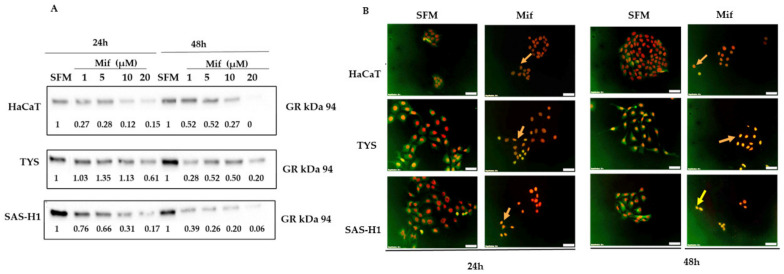
Effect of mifepristone (Mif) on the expression and localisation of the glucocorticoid receptor (GR) in HaCaT, TYS, and SAS-H1 cells. (**A**) Cells treated with 1, 5, 10, and 20 µM mifepristone were incubated for 24 and 48 h. Cells were then lysed, and the lysates were fractionated by SDS-PAGE using 10% acrylamide gels followed by transfer to PVDF membrane by Western blotting. The membrane was probed using primary antibody for GR (1:1000), followed by incubation with goat anti-rabbit secondary antibody (Table 1). The Western blots were normalised against total protein, quantified, and the data presented as fold-change in the GR expression compared to cells incubated in serum-free medium (SFM) only, at 24 and 48 h. A representative experiment is shown. Full blot images are presented in the Appendix A. (**B**) shows the merged immunofluorescence (IF) images for localisation of the GR in HaCaT, TYS, and SAS-H1 cells in response to mifepristone (Mif). Cells were treated with 20 µM mifepristone (Mif) and incubated for 48 h. Cells were then fixed and stained for GR using primary and fluorescent labelled secondary antibodies to analyse the localisation of the GR. Addition of mifepristone (Mif) resulted in complete nuclear translocation of the GR, as indicated by the arrows. The images were captured at 200× magnification using a fluorescence microscope (IX70) with a digital camera (XM10). Cells incubated with SFM were used as a control. Experiments were repeated three times. A representative experiment is shown.

**Figure 3 ijms-25-08777-f003:**
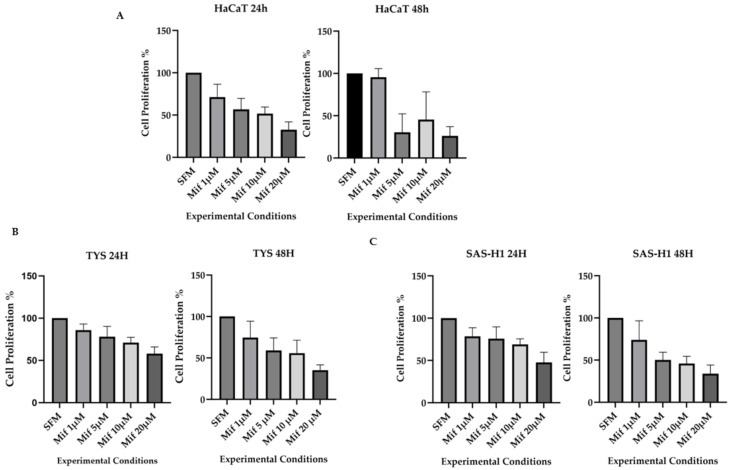
Effect of mifepristone (Mif) on the proliferation of HaCaT, TYS, and SAS-H1 cells. HaCaT (**A**), TYS (**B**), and SAS-H1 (**C**) cells were seeded in 48-well plates at a cell density of 2 × 10^4^ cells/mL. Following overnight attachment, cells were treated with mifepristone (Mif) at concentrations of 1, 5, 10, and 20 µM and incubated for 24 and 48 h. MTT solution, diluted in serum-free medium (SFM), was added to the cells and incubated for 3 h. The MTT was then removed, and DMSO was added. The plates were read on an OPTIMA plate reader at a wavelength of 550 nm with a reference filter of 620 nm. Data are expressed as the cell proliferation percentage compared to SFM, as a mean of four experiments, with error bars representing standard deviation (SD). Cells incubated in SFM were used as a control. A dose-dependent decrease in proliferation was observed in response to mifepristone for all the three cell lines.

**Figure 4 ijms-25-08777-f004:**
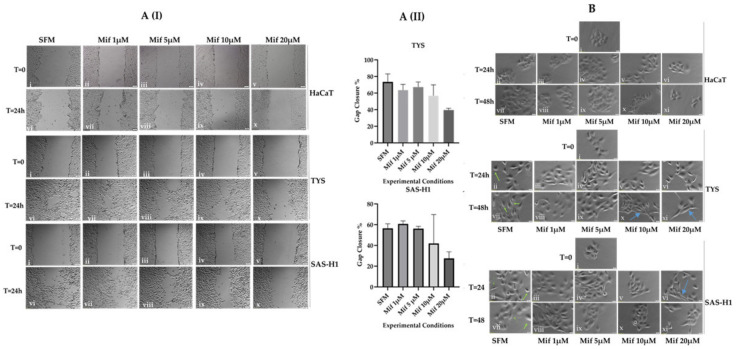
Effect of mifepristone on the migration of HaCaT, TYS, and SAS-H1 cell lines. **A** (I) (i–x) To observe the effect of mifepristone on collective cell migration, HaCaT, TYS and SAS-H1 cells were seeded at 5 × 10^5^ cells per 60 mm dish and incubated in 10% FCS-MEM until 90% confluency was achieved. A gap was then created in the monolayer. Cells were treated with mifepristone (Mif) concentrations of 1, 5, 10 and 20 µM and incubated for 24 h. Images were captured for all experimental conditions at time point 0 (T = 0) and after 24 h (T = 24 h), using the light microscope (IX70) and digital camera (XM10), at 40× magnification. (**A**) (II) The gap area at T = 0 and T = 24 h was analysed using Image J, and the data are presented as gap closure percentage, which decreased in response to increasing concentrations of mifepristone. Error bars represent the standard deviation (SD) in the three experiments. (**B**) (i–xi) To observe the effect of mifepristone on cell scatter, HaCaT, TYS, SAS-H1 cells were seeded at 4 × 10^(4)^ cells/mL and grown into small colonies. Cells were treated with mifepristone (Mif) concentrations of 1, 5, 10, and 20 µM for 24 and 48 h. After completion of the respective time points, cell scattering, and morphology were observed in response to treatment with varying concentrations of mifepristone (Mif). Images were taken at time points 0 (T = 0), 24 h (T = 24 h), and 48 h (T = 48 h) with a microscope (IX70) and digital camera (XM10) at 200×. Compacted colonies of TYS and SAS-H1 cells were observed in response to high concentrations (10 µM and 20 µM) of mifepristone. Cells incubated with serum-free medium (SFM) were used as a control. Green arrows indicate scattered cells in TYS and SAS-H1 control cultures. Blue arrows indicate elongated morphology in response to mifepristone. Experiments were repeated three times; a representative experiment is shown.

**Figure 5 ijms-25-08777-f005:**
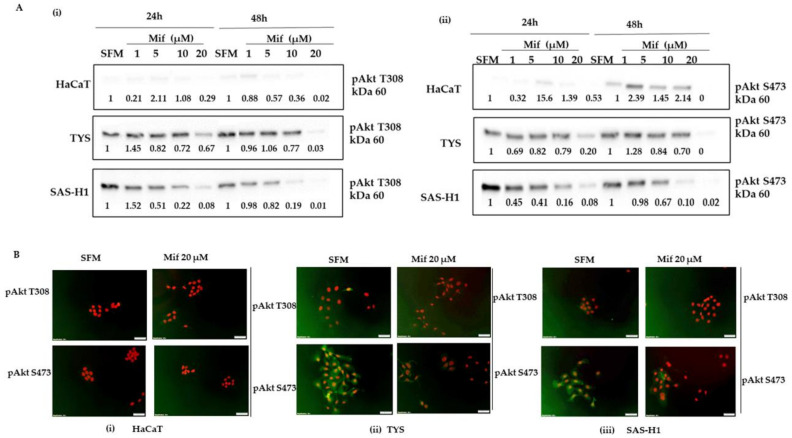
Effect of mifepristone (Mif) on levels of phosphorylated Akt Threonine 308 (pAkt T308) (**A**) (i), phosphorylated Akt Serine 473 (pAkt S473) (**A**) (ii), and their localisation (**B**), in HaCaT, TYS, and SAS-H1 cells at 24 and 48. Cells were treated with mifepristone (Mif) concentrations of 1, 5, 10, and 20 µM and incubated for 24 and 48 h. Cells were then lysed, and the lysates were fractionated by SDS-PAGE using 10% acrylamide gels followed by transfer to PVDF membrane by Western blotting. The membranes were probed using primary antibodies for pAkt T308 (1:1000) (**A**) (i) and pAkt S473 (1:2000) (**A**) (ii), followed by incubation with goat anti-rabbit secondary antibody (Table 1). The Western blots were normalised against total protein, quantified, and the data were presented as the fold-change in levels of pAkt T308 (**A**) (i) and pAkt S473 (**A**) (ii), compared to serum-free medium (SFM) at 24 and 48 h. Full blot images are presented in the Appendix A (Appendix A = pAkt 308, Appendix A = pAkt 473). (**B**) (i) HaCaT, (ii) TYS, and (iii) SAS-H1 were treated with 20 µM mifepristone (Mif) and incubated for 48 h. Cells were then fixed and stained for pAkt T308 and pAkt S473 using primary and fluorescent labelled secondary antibodies to analyse the localisation of pAkt T308 and pAkt S473. The images were captured using a fluorescence microscope (IX70) and digital camera (XM10) at 200× magnification. Cells incubated with SFM were used as a control. Experiments were repeated three times. A representative experiment is shown.

**Figure 6 ijms-25-08777-f006:**
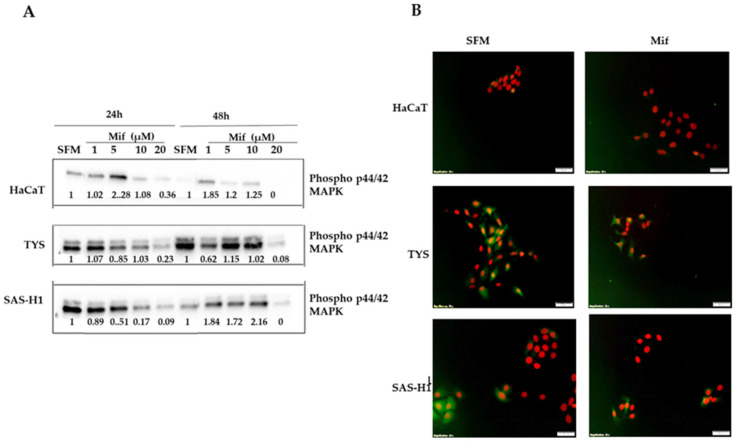
Effect of mifepristone (Mif) on levels of phosphorylated p44/42 mitogen-activated protein kinase (phospho p44/42 MAPK) (**A**), and its localisation (**B**), in HaCaT, TYS, and SAS-H1 cells at 24 and 48 h. (**A**) Cells were treated with mifepristone (Mif) concentrations of 1, 5, 10, and 20 µM and incubated for 24 and 48 h. Cells were then lysed, and the lysates were fractionated by SDS-PAGE using 10% acrylamide gels followed by transfer to a PVDF membrane by Western blotting. The membrane was probed using primary antibody for phospho p44/42 MAPK (1:2000), and goat anti-rabbit secondary antibody (Table 1). The Western blots were normalised against total protein, quantified, and the data were presented as the fold-change in levels of phospho p44/42 MAPK compared to serum-free medium (SFM) at 24 and 48 h. Full blot images are presented in the Appendix A. (**B**) HaCaT, TYS, and SAS-H1 were treated with 20 µM mifepristone (Mif) and incubated for 48 h. Cells were then fixed for immunofluorescence (IF) and stained for phospho p44/42 MAPK using primary and fluorescent labelled secondary antibodies to analyse the localisation of phospho p44/42 MAPK. The images were captured using a fluorescence microscope (IX70) and a digital camera (XM10) at 200× magnification. Cells incubated with SFM were used as a control. Experiments were repeated three times. A representative experiment is shown.

**Figure 7 ijms-25-08777-f007:**
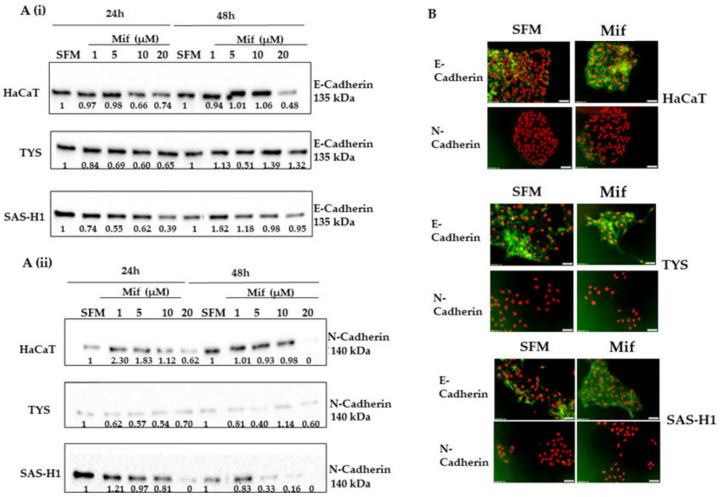
Effect of mifepristone (Mif) on the expression of E-Cadherin (**A**) (i), N-Cadherin (**A**) (ii), and their localisation (**B**), in HaCaT, TYS, and SAS-H1 cells at 24 and 48 h. Cells were treated with mifepristone (Mif) at concentrations of 1, 5, 10, and 20 µM and incubated for 24 and 48 h. Cells were then lysed, and the lysates were fractionated by SDS-PAGE using 10% acrylamide gels followed by transfer to a PVDF membrane by Western blotting. The membranes were probed using primary antibodies for E-Cadherin (1:1000) (**A**) (i), N-Cadherin (1:1000) (**A**) (ii), and goat anti-rabbit secondary antibody (Table 1). The Western blots were normalised against total protein, quantified, and the data are presented as the fold-change in levels of E-Cadherin (**A**) (i) and N-Cadherin (**A**) (ii) compared to serum-free medium (SFM), at 24 and 48 h. Full blot images are presented in the Appendix A. (**B**) HaCaT, TYS, and SAS-H1 were treated with 20 µM mifepristone (Mif) and incubated for 48 h. Cells were then fixed and stained for E-Cadherin and N-Cadherin using primary and fluorescent labelled secondary antibodies to analyse the localisation of E-Cadherin and N-Cadherin. The images were captured using a fluorescence microscope (IX70) and a digital camera (XM10) at 200× magnification. Cells incubated with SFM were used as a control. Experiments were repeated three times. A representative experiment is shown.

**Figure 8 ijms-25-08777-f008:**
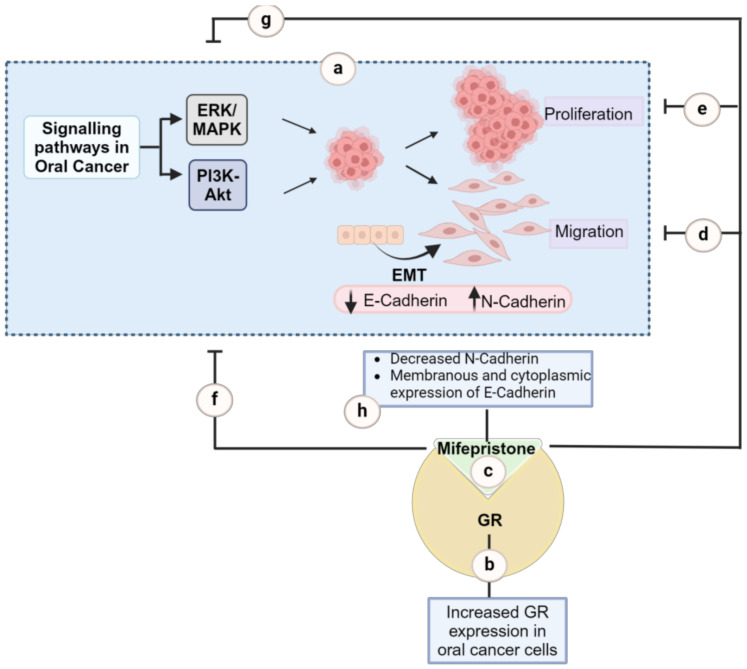
Summarizing the effect of mifepristone on the migration and proliferation of oral cancer cells. Cell proliferation and migration are the key hallmarks of cancer and are crucial to metastasis. Cell migration, is in turn, influenced by a transition in cellular morphology from epithelial to mesenchymal, a process known as epithelial–mesenchymal transition (EMT). Dysregulation of the PI3K-Akt and MAPK signalling pathways has been linked to increased cell proliferation and cell migration in HNC (a). The anticancer effects of glucocorticoid receptor antagonists, such as mifepristone, are widely reported for other cancer types with a lack of detailed exploration in oral cancer. This study bridges the gap by investigating the effects of mifepristone on cell migration, proliferation, and the underlying signalling pathways in oral cancer cells. Results showed increased expression of GR in oral cancer cells (b) and highlight the substantial impact of mifepristone (c), demonstrating that high concentrations (20 µM) effectively reduce the migration (d) and proliferation (e) of oral cancer cells. This effect is in accordance with the suppression of the Akt (f) and MAPK (g) signalling pathways, as well as the decreased expression of N-Cadherin (h). Despite the observed elongated cell morphology, which may be associated with the localisation pattern of E-Cadherin in response to mifepristone, the findings underscore the efficacy of mifepristone in inhibiting both the migration and proliferation of oral cancer cells. Further investigation is warranted to explore its impact on EMT markers to fully elucidate its role in HNC. Created with BioRender.com, accessed on 29 July 2024.

**Table 1 ijms-25-08777-t001:** Material and methods.

Materials	
Cell lines	Immortalised human skin keratinocytes (HaCaT) Oral adeno squamous cell carcinoma (TYS) Squamous cell carcinoma of the human tongue (SAS-H1)
Mifepristone	# M8046, sourced from Sigma-Aldrich. Reconstituted at 20 mg/mL in DMSO. Used in µM concentrations of 1, 5, 10, and 20 in serum-free media, (SFM).
Primary antibodies	pAkt S473 # 4060 (WB 1:2000, ICC 1:400), pAkt T308 #2965L (WB 1:1000, ICC 1:1600), p42/p44 pMAPK #4370 (WB 1:2000, ICC 1:400), Glucocorticoid receptor #12041 (WB 1:1000, ICC 1:200), E-Cadherin #3195 S (WB 1:1000, ICC 1:200), N-Cadherin #13116 S (WB 1:1000, ICC 1:800) All antibodies sourced from Cell Signaling Technology (Danvers, MA, USA)
Secondary antibodies	#7074 for Western blot, (1:2000) IgG Fab 2 Alexa Fluor for IF (1:1000) (Sourced from Cell Signaling Technology, Danvers, MA, USA)
DAPI (4′,6-diamidino-2-phenylindole)	# 4083, To stain cell nuclei in Immunocytochemistry (ICC), sourced from Cell Signaling Technology (Danvers, MA, USA)
	**Methods**
Cell migration	Scratch assay Scatter assay
Cell Proliferation	MTT assay ((3-(4,5-dimethylthiazol-2-yl)-2,5-diphenyl-2H-tetrazolium) assay (24 and 48 h)
Protein expression and localisation	Western blot Immunofluorescence (IF)
	**Software and Microscope**
Image Lab	To visualise and quantify the SDS-PAGE and Western blot image
Image J	To quantify the scratch assay
Photoshop	To merge DAPI and labelled antibody images
Microscope	Light and fluorescence microscope IX70; digital camera XM10

Details of cell lines, material, antibodies and their concentrations employed, assays, and software used in this study.

## Data Availability

Dataset available on request from the authors.

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
