# Peer review of "Effect of Mifepristone on Migration and Proliferation of Oral Cancer Cells"

_ijms, 2024, doi:10.3390/ijms25168777_

Round 1
Reviewer 1 Report
Comments and Suggestions for Authors
Dear Authors,
minor revisions are required.

Comments on the Quality of English LanguageAuthor Response
Please see attached file with reviewer comments and our answers.

Reviewer 2 Report
Comments and Suggestions for Authors
IJMS
Manuscript ID: ijms-3072780
Title: Effect of Mifepristone on Migration and Proliferation of Oral Cancer Cells
Special Issue: Head and Neck Cancer: From Molecular Diagnosis to Targeted Therapy
Date: 17 July 2024
1. Abstract
· The abstract mentions a range of assays (scratch and scatter assays, MTT assay, Western blot, immunocytochemistry) that are well-suited for studying cell proliferation, migration, and protein expression, indicating a rigorous approach.
· The abstract lacks specific details about the experimental conditions, such as the concentrations of mifepristone used and the duration of treatments. Providing these details would enhance the rigor by allowing for reproducibility and a better understanding of the experimental design.
· The abstract covers key aspects of the study, including the objective, methods, main findings, and conclusion, providing a well-rounded overview of the research.
· The language is clear and concise, making it easy to understand the research objective and findings.
· Appropriate scientific terminology is used, indicating the target audience is familiar with cancer biology and molecular techniques.
· Some terms, such as "PI3K-Akt and MAPK signaling pathways" and "EMT markers," might be too technical for a broader audience. Simplifying or briefly explaining these terms could improve accessibility without compromising accuracy.
· The conclusion succinctly summarizes the main findings, indicating that high concentrations of mifepristone suppress oral cancer cell migration and proliferation via specific signaling pathways.
· While it mentions the need for further investigation on EMT markers, it could be more specific about what aspects need further study or suggest potential applications of the findings.
· The keywords are relevant to the study and cover the main topics: glucocorticoid receptor, head and neck cancer, mifepristone, and oral cancer. Adding keywords related to the specific methodologies used (e.g., cell migration, proliferation assays) and signaling pathways (e.g., PI3K-Akt, MAPK) could improve the discoverability of the research.
2. Introduction
· The introduction provides a thorough background on mifepristone, covering its development, mechanism of action, and applications beyond its initial use.
· It touches on various applications of mifepristone in different cancer types and its potential as a therapeutic agent in oral cancer, demonstrating a wide range of considerations.
· While the background on mifepristone is comprehensive, the introduction could benefit from a more focused discussion on the specific context of oral cancer to set up the research question more directly.
· The introduction provides detailed information on the mechanisms of mifepristone, its historical and current uses, and its potential impact on cancer treatment.
· It identifies the gap in the literature regarding the role of mifepristone in cell migration in head and neck cancer (HNC), setting up the rationale for the study.
· The introduction could be more explicit about the hypothesis or specific research questions being addressed. While it mentions the investigation into migration and proliferation, a clear statement of hypothesis would enhance completeness.
· The language used is clear and precise, making complex biological concepts understandable.
· Appropriate use of scientific terminology enhances the professionalism and specificity of the text.
· Some sentences are long and complex, which could hinder readability. Breaking them into shorter sentences might improve clarity.
· The introduction logically concludes with the study's aims, effectively transitioning from the background information to the research objectives.
· The introduction references a range of studies, while the references are current, there is a need for a balanced mix of foundational and recent studies to provide a more comprehensive view.
· The literature review is extensive, covering various aspects of mifepristone's applications and mechanisms. It critically evaluates the complexities and challenges in the use of mifepristone, particularly in relation to its anti-cancer properties.
· The introduction clearly identifies a gap in knowledge regarding the role of mifepristone in cell migration in HNC.
· Include a clear and explicit statement of the hypothesis or research questions.
· Simplify complex sentences to enhance readability and clarity.
· Provide a stronger call to action or highlight the importance of the study's potential impact in the conclusion.
· The molecular mechanisms underlying mifepristone's effects on GR localization and expression are not fully elucidated.
· The long-term effects of mifepristone treatment on oral cancer cells are not explored.
· The potential combinatory effects of mifepristone with other therapeutic agents are not investigated
Results
· The study demonstrates methodological rigor through multiple experimental replicates and detailed protocols for Western blotting, immunofluorescence, cell proliferation assays, and scratch assays. However, some areas lack completeness:
· Control experiments are well-described, but there is limited discussion on the variability between replicates.
· The language is clear and technical, appropriate for a scientific audience. However, the presentation could be improved.
· Providing more detailed figure legends and integrating supplementary figures within the main text for easier reference.
· · Improve the clarity of figures and legends, and integrate supplementary materials more effectively within the main text.
Discussion
· The discussion section presents a thorough analysis of the results and compares them with findings from previous studies. The discussion rigorously addresses the effects of mifepristone on oral cancer cells, detailing its influence on cell migration, proliferation, and signaling pathways. The section effectively correlates the study's findings with existing literature, highlighting both consistencies and discrepancies.
· The language used in the discussion is clear and precise, making complex scientific concepts accessible. However, some sentences are lengthy and could benefit from being broken down for better readability. Additionally, minor grammatical improvements could enhance the overall clarity and flow of the text.
· The discussion section extensively cites recent literature, ensuring that the findings are contextualized within the current scientific landscape. References to studies from various cancer types, such as breast, ovarian, and prostate cancers, provide a comprehensive understanding of mifepristone's effects across different contexts.
· The discussion effectively compares the study's findings with existing literature. For instance, the observation of mifepristone-induced nuclear translocation of GR is consistent with previous reports in other cancer cell lines. However, the discussion also identifies discrepancies, such as the elongated morphology of cells treated with mifepristone, which contrasts with the rounded phenotype observed in other studies. These comparisons are crucial for highlighting the unique aspects of the study and identifying areas for further research.
· The conclusion is well-aligned with the data presented and accurately reflects the study's contributions to the field. The suggestion for further investigation into the impact of mifepristone on EMT markers and related signaling pathways is appropriate and highlights the need for additional research to fully elucidate mifepristone's role as an anti-cancer agent in head and neck cancer.
· Simplify complex sentences to improve readability. For example, breaking down lengthy sentences into shorter, more concise statements can enhance clarity.
· Review the text for minor grammatical errors to improve overall readability.
· While the discussion provides a comprehensive overview, adding more detailed mechanistic insights into how mifepristone affects the PI3K-Akt and MAPK signaling pathways could further strengthen the discussion.
· Expand the comparison with literature to include more studies on head and neck cancer, as this would provide a more focused context for the findings.
· Consider incorporating visual aids, such as diagrams or flowcharts, to illustrate the key signaling pathways affected by mifepristone. This could help readers better understand the complex interactions described.
- The discussion on mifepristone's agonistic properties at low doses could be expanded to provide a clearer understanding of the conditions under which these properties are observed.
- More detailed discussion on the specific roles of the PI3K-Akt and MAPK signaling pathways in the context of oral cancer could strengthen the connection between the experimental findings and their biological implications.
- Explicitly stating the limitations of the study, such as the use of specific cell lines or the range of concentrations tested, would provide a more comprehensive view of the research's scope and applicability.
- Offering more specific suggestions for future research, particularly in investigating additional EMT markers and related signaling pathways, would guide subsequent studies in this area.
· The conclusion drawn is suitable and supported by the data presented. It accurately summarizes the findings and their implications, particularly the potential of mifepristone as an anti-cancer agent in HNC. However, it also appropriately notes the need for further investigation, especially regarding the cytoplasmic localization of E-Cadherin and its impact on cell behavior.
Comments on the Quality of English LanguageMinor editing of English language required
Author Response
Please see attached file containing reviewer comments and answers.

Reviewer 3 Report
Comments and Suggestions for Authors
The use of mifepristone itself is not innovative, the use of this medicine dates back well over ten years. Patients with the response or stable disease could continue with the same blinded therapy. Patients who progressed on placebo could be switched to treatment with mifepristone. The primary endpoint was failure-free survival. In conclusion, long-term administration of mifepristone was well tolerated but had no impact on patients with unresectable meningioma.
The innovative part could reside in the use of oral cell carcinoma...
However, consider two things the authors could take and use in their paper.
1 In humans, PR is expressed in the stroma of the prostate. Substantially, a lower PR expression in cancer-associated stroma may eventually have an effect away from the original affected area, it may promote the creation of a tumor microenvironment conducive to tumor cell invasion and metastasis. Thus, if the presence of PR may block tumor invasion and growth, treatment with a PR antagonist may worsen the condition somewhere else...
2 Though this is an in vitro study, there are still engineered mice for HNC, why did the authors not consider developing an in vivo study? It is well known that the microenvironmental factors that may contribute to HNC, HPV, and EBV play a critical role....
3 Many points and topics should be reconsidered and added in this article
Comments on the Quality of English Language
English does not show a big problem
Author Response
Please see attached file with reviewer comments and answers.

Round 2
Reviewer 3 Report
Comments and Suggestions for Authors
For me, this paper can be accepted